# U-Net Inspired Transformer Architecture for Multivariate Time Series Synthesis

**DOI:** 10.3390/s25134073

**Published:** 2025-06-30

**Authors:** Shyr-Long Jeng

**Affiliations:** Department of Mechanical Engineering, Lunghwa University of Science and Technology, Taoyuan City 333326, Taiwan; aetsl@gm.lhu.edu.tw; Tel.: +886-2-8209-3211 (ext. 5113)

**Keywords:** attention, CLLC converter, half-bridge, time series synthesis, electric vehicle

## Abstract

This study introduces a Multiscale Dual-Attention U-Net (TS-MSDA U-Net) model for long-term time series synthesis. By integrating multiscale temporal feature extraction and dual-attention mechanisms into the U-Net backbone, the model captures complex temporal dependencies more effectively. The model was evaluated in two distinct applications. In the first, using multivariate datasets from 70 real-world electric vehicle (EV) trips, TS-MSDA U-Net achieved a mean absolute error below 1% across key parameters, including battery state of charge, voltage, acceleration, and torque—representing a two-fold improvement over the baseline TS-p2pGAN. While dual-attention modules provided only modest gains over the basic U-Net, the multiscale design enhanced overall performance. In the second application, the model was used to reconstruct high-resolution signals from low-speed analog-to-digital converter data in a prototype resonant CLLC half-bridge converter. TS-MSDA U-Net successfully learned nonlinear mappings and improved signal resolution by a factor of 36, outperforming the basic U-Net, which failed to recover essential waveform details. These results underscore the effectiveness of transformer-inspired U-Net architectures for high-fidelity multivariate time series modeling in both EV analytics and power electronics.

## 1. Introduction

Time series data, consisting of sequential observations collected at regular intervals, are fundamental to various industrial applications, including electric vehicle (EV) battery monitoring for state of charge (SOC) estimation and high-resolution signal reconstruction in high-frequency resonant CLLC half-bridge converters. Time series synthesis (TSS) plays a critical role for generating synthetic sequences that accurately replicate the statistical properties and temporal dependencies of real-world data. This technique is vital for augmenting limited datasets, enhancing model generalization while maintaining intrinsic trends and variability [1]. With the increasing prevalence of time series data in engineering and cyber-physical systems [2], TSS has become indispensable for building robust models across diverse domains.

Generative adversarial networks (GANs) have demonstrated considerable success in TSS by training a generator to produce synthetic sequences and a discriminator to distinguish them from real data [3]. Models such as TimeGAN [4] excel in capturing complex temporal dependencies, particularly for energy consumption data augmentation. Hybrid GAN-based models further extend these capabilities. For example, CR-GAN [5] integrates convolutional neural networks (CNNs) and long short-term memory (LSTM) networks to jointly extract both spatial and temporal features. Recent advancements include deep convolutional GAN frameworks [6] for enhancing sparse lithium-ion battery datasets while preserving dynamic behavior and TS-p2pGAN [7], which adapts image-based pixel-to-pixel GAN frameworks to synthesize long-sequence EV data under realistic driving scenarios.

Traditional CNNs are limited by their local receptive fields, restricting their ability to capture long-range dependencies critical for modeling complex temporal patterns [8]. To mitigate this, strategies such as dilated convolutions, pooling-based multiscale feature aggregation, and larger kernel sizes have been proposed. These methods often struggle in capturing diverse multiscale features, leading in suboptimal performance, particularly when segmenting objects or sequences with varying structures. Geometric awareness has recently been introduced into neural networks [9,10], enhancing the accuracy and robustness of point detection by incorporating spatial and structural context. Although techniques like atrous (dilated) convolutions [11] can expand receptive fields, they remain fundamentally local in nature and are limited in modeling global contextual relationships. The U-Net architecture [12], initially designed for medical image segmentation, has demonstrated adaptable for time series modeling. However, traditional encoder–decoder frameworks often struggle with noise introduced by defective modalities. To address such challenges, ConTriNet [13] adopts a divide-and-conquer strategy inspired by the hierarchical structure of the human visual system, enhancing robustness and accuracy in salient object detection across diverse scenarios. Its encoder–decoder structure constructs hierarchical representations while preserving critical features through skip connections [14,15]. Despite its strengths, basic U-Net remains constrained in handling highly complex spatial or temporal dynamics, motivating the need for architectural enhancements.

Transformers, first introduced for natural language processing [16], have increasingly been applied to time series tasks due to their self-attention mechanism, which effectively model long-range dependencies. Attention-based models [17] significantly improve forecasting accuracy by identifying temporal correlations. However, conventional attention modules operate pointwise, often underutilizing broader temporal structures. Hybrid approaches combining CNNs with transformers [18,19] have improved tasks like image super-resolution, where transformer encoders extract global features from CNN maps, and decoders refine details via cross-attention mechanisms. A recent advancement, FedTime [20], introduces a federated learning framework using a pre-trained Large Language Model (LLM) for long-term forecasting. Particularly suited for energy demand prediction, FedTime enables accurate, privacy-preserving forecasting across decentralized datasets, highlighting the growing relevance of scalable and communication-efficient time series models in energy systems.

In medical image segmentation, several U-Net-based architectures with transformer mechanisms have emerged, each offering unique advantages and tackling specific computational challenges. TransUNet [21] is a hybrid model that combines a CNN backbone with a vision transformer in its bottleneck stage. This allows TransUNet to leverage the CNN’s strength in extracting low and mid-level spatial features while benefiting from the transformer’s ability to model long-range dependencies through global self-attention. However, its use of a standard vision transformer can lead to high computational costs due to the non-hierarchical nature of global attention.

The Swin-Unet model [22] integrates the Swin Transformer, a hierarchical vision transformer with shifted windows, into the U-Net framework. This design enables efficient local and global context modeling with linear computational complexity. Its multiscale capability is well suited for varying spatial resolutions. However, its window-based attention may struggle with very large or irregular contexts unless supported by deeper networks.

UNETR [23] adopts a different approach by replacing the convolutional encoder with a vision transformer while retaining the U-Net decoder and skip connections. This configuration enables early-stage global dependency modeling and achieves high performance in complex segmentation tasks, though at the cost of increased memory and computational demands due to the quadratic complexity of self-attention.

UNETR++ [24] enhances UNETR with efficient paired attention blocks to learn spatial-wise and channel-wise discriminative features for 3D segmentation. Architectural improvements such as deep supervision, multiscale fusion, and refined skip connections improve feature integration and gradient flow. While UNETR++ often outperforms Swin-Unet and UNETR, it introduces additional complexity, longer training times, and higher hyperparameter sensitivity, limiting its practicality for datasets with limited annotations.

Despite the progress in U-Net variants, many models emphasize spatial attention while underutilizing channel information and inter-feature dependencies. This gap can hinder segmentation quality, especially in tasks requiring rich contextual representation. Addressing both spatial-wise and channel-wise interactions is crucial for more accurate predictions. The U-Net architecture has gained traction in TSS for its ability to capture multiscale features while preserving temporal detail. As research advances, adapting U-Net frameworks for TSS presents promising opportunities for modeling complex temporal structures across multiple scales, offering significant potential for applications in engineering systems and beyond.

The main contributions of this work are summarized as follows:Dual-attention (DA) Mechanisms in U-Net Framework: The proposed TS-MSDA U-Net model integrates a hierarchical encoder–decoder structure for multiscale temporal feature extraction with DA mechanisms, comprising both sequence attention (SA) and channel attention (CA), effectively capturing complex temporal dynamics in multivariate time series data.Enhanced TSS for EVs: The proposed model achieves low mean absolute errors (MAEs), all within 1% of ground truth values, across key EV parameters (battery SOC, voltage, acceleration, and torque) using an open-source dataset from 70 real-world trips. Compared to the baseline TS-p2pGAN model, it yields a two-fold reduction in MAE.High-Resolution Signal Reconstruction: The TS-MSDA U-Net achieves a 36× enhancement in signal resolution from low-speed ADC data of a resonant CLLC half-bridge converter, successfully capturing complex nonlinear mappings where the basic U-Net models failed.Cross-Domain Validation and Attention Mechanism Analysis: The model is validated across two distinct engineering domains: automotive and power electronics, demonstrating generalizability. In the automotive domain, the baseline U-Net already achieves strong performance over TS-p2pGAN, and the addition of the DA mechanism yields a modest improvement of approximately 0.2–0.3% in MAE and RMSE metrics. Conversely, for high-frequency signal reconstruction in the resonant CLLC converter, the DA module is essential: The basic U-Net fails to capture waveform details, while the DA-enhanced model achieves successful reconstruction. This contrast highlights the DA module’s critical role in tasks requiring fine-grained temporal–spatial representation and provides insight into its domain-dependent effectiveness.

The remainder of this paper is organized as follows. Section 2 introduces the proposed hierarchical encoder–decoder architecture tailored for multiscale temporal learning. It also details DA block, comprising SA and CA modules, which enhance the model’s ability to capture complex temporal and inter-feature dependencies in multivariate time series data. Section 3 presents the experimental results, structured into two parts. The first part focuses on the EV trip dataset, an open-source dataset containing 70 real-world EV driving sessions, and compares the proposed model’s performance with the TS-p2pGAN baseline. The second part examines periodic signal reconstruction in a resonant CLLC half-bridge converter. This includes the generation of time series training data using the PLECS simulator, training evaluation, and testing outcomes from prototype hardware. Section 4 concludes the paper by summarizing the major findings and contributions.

## 2. Materials and Methods

### 2.1. Hierarchical Encoder–Decoder Network

The TS-MSDA U-Net architecture, illustrated in Figure 1, is designed for TSS through a hierarchical encoder–decoder network enhanced with dual-attention–convolution (DA-conv) blocks. In contrast to the UNETR model [23], which replaces convolutional encoders with transformers, TS-MSDA U-Net preserves the basic U-Net’s multilevel structure, facilitating effective multiscale feature extraction.

The encoder consists of four stages that progressively downsample the input time series data. This is accomplished using a DA-conv block followed by one-dimensional max pooling. The input sequence $X\in \;R^{L\times C_{in}}$, where L represents the sequence length and $C_{in}$ the input channel dimension, is configured with (L = 512, $C_{in}$ = 10) for real-world EV driving data and (L = 512, $C_{in}$ = 6) for resonant CLLC half-bridge converter data. In the first encoder stage, a DA-conv block projects the channel dimension from $C_{in}$ to 64 channels. Subsequently, each encoder stage applies downsampling through 1D max pooling, halving the sequence length. Simultaneously, DA-conv blocks double the number of feature channels at each stage. The encoding process culminates in a bottleneck layer that applies a final DA-conv block, producing a feature map of size 32 × 1024, which is forwarded to the decoder.

The decoder mirrors the encoder’s structure with four symmetric stages. Each stage begins with a one-dimensional deconvolution layer that upsamples the feature map, doubling both the temporal resolution and the number of channels. Skip connections connect each encoder stage to its corresponding decoder stage, allowing their outputs to be concatenated and passed through a DA-conv block. At each decoder stage, the number of feature channels is halved to maintain symmetry with the encoder. The final decoder output is merged with earlier convolutional features to restore sequence-level detail and refine the representation. A concluding 1 × 1 convolutional layer then produces the synthesized time series output with dimensions 512 ×$\;C_{out}$, where $C_{out}$ is the desired number of output channels.

The DA-conv block, a key component of the architecture, integrates a dual-attention (DA) module followed by a convolutional block. The DA module captures dependencies along both the sequence and channel dimensions using a key–query attention mechanism. This DA enhances the model’s ability to learn joint temporal and inter-channel patterns. The subsequent convolutional block consists of a 1D convolutional layer, batch normalization, and a ReLU activation function. This combination allows the DA-conv block to efficiently model complex temporal relationships in multivariate time series data.

A notable strength of TS-MSDA U-Net is the use of high channel dimensionality in the decoder’s upsampling path, which facilitates the preservation and propagation of contextual information to higher-resolution layers. The combination of alternating downsampling and upsampling, enriched by skip connections, enables deep hierarchical feature learning and robust time series reconstruction. The final convolutional layer effectively integrates these features to generate high-fidelity synthetic sequences, recovering information lost during encoding and improving the precision of the output. The full layer-wise architecture and associated hyperparameters are detailed in Table 1.

### 2.2. Dual-Attention Block

The dual-attention (DA) block, depicted in Figure 2, is a core component embedded within each DA-conv block across both the encoder (contracting) and decoder (expanding) paths of the TS-MSDA U-Net. This block combines a SA module and a CA module to jointly enhance temporal and inter-channel feature modeling. To ensure the attention mechanisms are aware of positional information, positional embeddings are introduced prior to the attention layers.

#### 2.2.1. Learned Positional Embedding

Unlike fixed schemes such as sinusoidal embeddings, learned positional embeddings introduce positional information through trainable vectors. Each position in a time series sequence is associated with a unique vector that is updated during training. Given a time series input $X\in R^{L\times C}$, where L is the sequence length and C is the channel dimension, a positional embedding matrix $P\in R^{L\times C}$ is added to the input:(1)$$X^{\prime }=X+P$$
where $X^{\prime }$ contains both the raw time series features and learned positional context, serving as input to the DA modules. This formulation provides the model with both raw features and temporal context, which is passed to the downstream dual-attention modules. Its empirical advantage in capturing task-specific temporal patterns without being limited by the constraints of fixed embedding schemes.

#### 2.2.2. Sequence Attention Module

The SA module is designed to model long-range sequence dependencies along the temporal axis. It operates on the transposed input feature matrix ${\left(X^{\prime }\right)}^{T}\in R^{C\times L}$, effectively treating each channel as a token sequence. From this transposed input, the query ($Q_{s}^{*}$), key ($K_{s}^{*}$), and value ($V_{s}$) matrices are generated via learned linear projections:(2)$$Q_{s}^{*}=W^{Q_{s}}\cdot {\left(X^{\prime }\right)}^{T},\;K_{s}^{*}=W^{K_{s}}\cdot {\left(X^{\prime }\right)}^{T},\;\mathrm{and}\;\;\;V_{s}=W^{V_{s}}\cdot {\left(X^{\prime }\right)}^{T}$$
where $W^{Q_{s}}$, $W^{K_{s}}$, and $W^{V_{s}}\in R^{C\times C}$ represent learnable weight matrices. The sequence attention map $A_{s}\in R^{L\times L}$ is computed by measuring the similarity between temporal positions:(3)$$A_{s}=\mathrm{softmax}\left({\left(K_{s}^{*}\right)}^{T}\cdot Q_{s}^{*}\right)\;$$

This attention map captures dependencies across time steps. The output of the SA module is then calculated as(4)$${\left(O_{s}\right)}^{T}=V_{s}\cdot A_{s}$$
where $O_{s}^{T}\in R^{C\times L}$ is subsequently transposed to match the original input shape.

#### 2.2.3. Channel Attention Module

The CA module operates in parallel with the SA module but focuses on capturing dependencies along the channel dimension. Using the original input $X^{\prime }\in R^{L\times C}$, the module computes the query, key, and value projections,(5)$$Q_{c}^{*}=W^{Q_{c}}\cdot X^{\prime },\;K_{c}^{*}=W^{K_{c}}\cdot X^{\prime },\;\mathrm{and}\;V_{c}=W^{V_{c}}\cdot X^{\prime }$$
where $W^{K_{c}}$, $W^{K_{c}}$, and $W^{V_{c}}\in R^{L\times L}$ are learnable parameters. The channel attention map $A_{c}$ $\in R^{C\times C}$ is obtained by(6)$$A_{c}=\mathrm{softmax}\left({\left(K_{c}^{*}\right)}^{T}\cdot Q_{c}^{*}\right)\;$$
and the output of the CA module is computed as(7)$$O_{c}=V_{c}\cdot A_{c}\;$$

This process allows the CA module to identify salient inter-channel relationships.

#### 2.2.4. Shared Query/Key Projections and Feature Fusion

Given that, in most time series applications, the sequence length L significantly exceeds the number of channels C, the computational burden of computing large attention maps becomes a concern. To reduce complexity and promote shared learning, the TS-MSDA U-Net reuses query and key projections from the SA module in the CA module as follows:(8)$$Q_{c}^{*}={\left(K_{s}^{*}\right)}^{T}\;\mathrm{and}\;K_{c}^{*}={\left(Q_{s}^{*}\right)}^{T}$$

Using this shared representation, the channel attention map can be equivalently expressed as(9)$$A_{c}=\mathrm{softmax}\left({\left(K_{c}^{*}\right)}^{T}\cdot Q_{c}^{*}\right)=\mathrm{softmax}\left(Q_{s}^{*}\cdot {\left(K_{s}^{*}\right)}^{T}\right)$$
where $A_{c}$ $\in R^{C\times C}$ remains consistent in capturing inter-channel dependencies.

Finally, the outputs of the SA and CA modules are concatenated along the channel dimension to form the final output of the DA block. This fused representation integrates both temporal and channel information, producing a richer and more context-aware feature map for TSS.

## 3. Experimental Setup and Results

To evaluate the effectiveness of U-Net variants in synthesizing long-term time series data, two distinct experiments were conducted.

The first experiment focused on synthesizing four critical driving features extracted from a dataset comprising 70 EV trips conducted under a range of real-world driving conditions. The performance of the proposed TS-MSDA U-Net was rigorously benchmarked against several baseline and advanced models, including the basic U-Net [12], U-Net with SA, UNETR [23], UNETR++ [24], and previously published results from the TS-p2pGAN model [7]. These comparisons provided a comprehensive assessment of the proposed model’s performance in realistic EV scenarios.

The second experiment investigated the enhancement of ADC signal resolution. Interestingly, this experiment revealed that, while attention mechanisms offered only marginal improvements over the basic U-Net in relatively simple synthesis tasks, they significantly enhanced the fidelity of generated outputs in more complex, nonlinear scenarios. This highlights the value of attention-based modeling in capturing intricate sequence-wise and channel-wise dependencies.

All models were implemented in Python (version 3.11.6)) using the PyTorch (version 2.3.1) deep learning framework. Experiments were executed on a system equipped with an Intel Core i7-10700K CPU and an NVIDIA GeForce RTX 4090 GPU with 24 GB of VRAM. The datasets used in the first and second experiments consisted of 70,090 and 79,872 segments, respectively. For both tasks, the data were randomly partitioned into 80% training and 20% validation sets.

Model training was conducted using the Adam optimizer, with the MAE as the objective loss function. In Experiment 1 (automotive domain), the model was trained for 200 epochs with an initial learning rate of 2 × 10⁻⁴. This rate was held constant for the first 150 epochs, then linearly decayed to zero over the remaining 50 epochs. In Experiment 2 (power electronics domain), the model was trained for 120 epochs with an initial learning rate of 6 × 10⁻⁴, which remained fixed for the first 80 epochs before linearly decaying to zero during the final 40 epochs. A consistent batch size of 256 was used across all training runs. To ensure training stability and reduce the risk of overfitting, the datasets were fully reshuffled at the beginning of each epoch. Training and validation losses were carefully monitored to assess generalization performance. To quantitatively evaluate the similarity between the real and synthesized time series, three performance metrics were employed: Root Mean Square Error (RMSE), MAE, and Dynamic Time Warping (DTW). The DTW metric [25] was particularly important for capturing temporal misalignments between the real and synthesized sequences. By accounting for nonlinear time distortions, DTW offered a more robust and informative assessment of sequence-level similarity compared to point-wise metrics alone.

### 3.1. Vehcile Trip Dataset

A comprehensive dataset focused on high-voltage battery behavior in EVs was obtained from the IEEE DataPort repository [26]. This dataset comprises real-world driving data collected from 70 distinct trips taken by a BMW i3 (60 Ah) under both summer (Group A) and winter (Group B) conditions. These varied environmental settings provide valuable insights into EV and battery performances under diverse operational and climatic scenarios.

To ensure data quality, all time series were checked for missing values, and sporadic gaps were filled using linear interpolation from adjacent valid entries. Each trip includes 30 recorded variables, capturing a wide spectrum of information such as environmental conditions, vehicle dynamics, battery performance metrics, and details of the heating system. For training the TS-MSDA U-Net model, 10 key variables were selected based on their relevance and perceived impact on EV battery behavior and overall system dynamics. These include vehicle speed, altitude, throttle position, motor torque, acceleration, battery voltage, battery current, battery temperature, state of charge (SOC), displayed SOC, power consumption of the heater and air conditioner, heater voltage, heater current, and ambient temperature.

The raw dataset, originally sampled at 100-millisecond intervals, was first preprocessed using a 1-s moving average filter, effectively smoothing short-term fluctuations by averaging every ten consecutive samples. The resulting smoothed time series data was then segmented into overlapping sequences, each containing 512 consecutive data points, using a sliding window approach with a stride of 1. This stride of 1 implies a high degree of window overlap, specifically 511 data points of overlap between consecutive segments, thereby ensuring continuous and comprehensive temporal coverage and maximizing the utilization of the available data. To standardize the input and facilitate stable model training, all time series sequences were normalized to a [−1, 1] range.

#### Baseline Comparison

Figure 3 presents the training and validation loss curves, measured by MAE, for five model variants. As shown in Figure 3a, all models demonstrated a steady reduction in training loss, indicating effective convergence. To better highlight subtle differences—particularly during the plateauing stages where absolute differences in loss are small—a dual y-axis with the right axis displaying a logarithmic scale was used. While the baseline U-Net displayed solid overall performance, the TS-MSDA U-Net and UNETR++ through its synergistic integration of multiscale dual-attention and transformer-inspired components, achieved slightly lower training losses, indicating enhanced optimization and learning capability. In contrast, U-Net with SA and UNETR showed relatively higher training losses, possibly due to their increased architectural complexity or greater sensitivity to the data’s temporal characteristics.

Figure 3b shows the corresponding validation loss curves, which generally mirrored the training trends, demonstrating good generalization across models. Similar to Figure 3a, the right y-axis is presented on a logarithmic scale for enhanced visualization of subtle differences at lower loss values. However, with the exception of TS-MSDA U-Net, the other occasionally exhibited spikes in validation loss. These transient fluctuations may be due to the transformer-based models’ sensitivity to sequence variability or abrupt changes in input dynamics or to the U-Net’s responses to outlier patterns. Notably, these spikes were often followed by recovery in the next epoch, suggesting a degree of model resilience and adaptation. Despite these fluctuations, the overall model performance remained stable. Importantly, the proposed TS-MSDA U-Net consistently achieved the lowest and most stable validation loss throughout training. This stability and predictive accuracy, particularly when other complex models exhibited transient fluctuations, serve as evidence of the effectiveness and integration within our architecture, underscoring its feature learning and reliable generalization compared to the other model variants.

Figure 4 illustrates the performance of the proposed TS-MSDA U-Net in synthesizing four critical driving parameters—SOC, battery voltage, motor torque, and longitudinal acceleration—throughout the entire 3240-s duration of the B01 trip. A visual comparison between the synthetic signals (shown in blue) and the corresponding real-world measurements (in red) reveals a high degree of alignment. The green dotted lines, representing the error margins between the synthetic and real signals, remain consistently narrow and well-contained within predefined bounds.

To ensure a fair comparison with the previously reported TS-p2pGAN model [7], the upper and lower limits of the error bands were set to match the reference ranges used in that study. Two magnified time intervals—specifically [648, 904] and [1942, 2198]—are highlighted to examine the model’s ability to replicate rapid transitions and short-term fluctuations typical of real-world driving conditions. Within these windows, the TS-MSDA U-Net effectively captures high-frequency variations and dynamic changes with minimal lag or distortion.

While minor discrepancies were observed in the synthesized motor torque and longitudinal acceleration, particularly at sharp peaks or during rapid transitions, these deviations were generally small. Crucially, the TS-MSDA U-Net exhibited reduced error magnitudes compared to the TS-p2pGAN, which previously struggled with modeling high-frequency components. The TS-MSDA U-Net maintained error distributions tightly centered around zero, indicating a high degree of temporal fidelity and generalization capability in replicating complex EV behaviors.

Table 2 provides a detailed comparative analysis of the five U-Net model variants alongside the previously proposed TS-p2pGAN, evaluated across all driving trips using three key performance metrics: RMSE, MAE, and DTW. The TS-MSDA U-Net consistently demonstrated strong performance, frequently achieving the lowest RMSE, MAE, and DTW values across a majority of driving scenarios. For example, in Trip 1, the TS-MSDA U-Net achieved an RMSE of 0.68%, an MAE of 0.39%, and a DTW of 0.47%, while TS-p2pGAN recorded substantially higher errors of 1.96%, 0.97%, and 1.19%, respectively. These results illustrate the model’s effectiveness in capturing EV battery dynamics under realistic driving conditions.

Compared to the baseline U-Net, neither the U-Net with SA nor the UNETR model demonstrated consistent improvements. In some cases, such as Trip 41, UNETR recorded the highest error values among all U-Net variants, with an RMSE of 5.57%, an MAE of 2.94%, and a DTW of 3.34%. This suggests that its emphasis on modeling long-range dependencies may not effectively capture the localized temporal and channel-level variations present in EV time series data. In contrast, TS-MSDA U-Net’s integration of multiscale feature extraction with dual-attention mechanisms offers more flexibility for modeling such variations.

To further support these empirical observations, statistical significance testing was conducted using two-tailed paired *t*-tests on RMSE, MAE, and DTW values across all trips. As shown in Table 3, TS-MSDA U-Net exhibits statistically significant improvements over U-Net, U-Net with SA, UNETR, and TS-p2pGAN on all three metrics, with *p*-values well below the 0.05 threshold. This confirms that the performance gains are not due to chance but are statistically reliable. When compared to UNETR++, the results are more nuanced: While the differences in RMSE (*p* = 8.22 × 10^−8^) and DTW (*p* = 2.18 × 10^−4^) remain statistically significant, the MAE difference (*p* = 0.448) is not. This suggests that although TS-MSDA U-Net consistently achieves better overall reconstruction accuracy and alignment, some localized error patterns are comparable between the two models.

UNETR++ showed modest improvements over its predecessor, occasionally achieving performance comparable to that of TS-MSDA U-Net on specific trips. However, across the full dataset, TS-MSDA U-Net demonstrated more stable error distributions and lower variability in metrics, indicating better generalization across varied scenarios. For instance, the standard deviation of its RMSE values across all trips was consistently lower than those of other attention-based variants, highlighting its robustness rather than asserting absolute superiority. Among all the models, TS-p2pGAN exhibited the most inconsistent performance, with relatively high errors across numerous trips, pointing to its limited generalization capacity in diverse driving conditions. These comparative findings support the practical effectiveness of TS-MSDA U-Net, particularly in terms of reliability and adaptability.

Figure 5 presents a visual comparison of the discrepancies in four key synthesized features between the TS-MSDA U-Net and the previously established TS-p2pGAN models across all driving trips, utilizing violin plots to provide a comprehensive view of error distributions. The input segment length differed between the two models. TS-MSDA U-Net used 512-sample sequences, while TS-p2pGAN used 256-sample sequences. As a result, Trips 13, 34, and 42 were excluded from the TS-MSDA U-Net evaluation due to insufficient duration.

Trained with the MAE loss metric, the TS-MSDA U-Net produced discrepancy distributions with centroids closely centered around zero across all features, suggesting a more balanced error profile and improved predictive consistency. The violin plots visually capture these patterns by illustrating the full distributional structure—including density, spread, and symmetry—of each model’s errors. In contrast, TS-p2pGAN displayed broader and more asymmetric error distributions, reflecting greater variability in prediction quality across trips.

More specifically, the TS-MSDA U-Net demonstrated tightly concentrated error distributions, with mean absolute errors (MAEs) below 1% across all four features, underscoring its high precision and stability across different trips. By comparison, TS-p2pGAN showed greater variability, particularly in motor torque and longitudinal acceleration, where discrepancies occasionally reached up to 5%, indicating a broader error distribution and reduced prediction fidelity.

The shape of the discrepancy distributions further highlighted the TS-MSDA U-Net’s consistency. For most trips, the model produced symmetrical, bell-shaped curves, indicative of a well-balanced and normally distributed error profile. A few exceptions were observed, for example, Trips 4, 14, 33, 41, and 62 for SOC and Trip 41 for battery voltage, which the distributions showed slight asymmetry or broader spread. These trips are marked with red dashed boxes in Figure 5a to indicate deviations from the typical pattern. In contrast, the TS-p2pGAN model more frequently deviated from this ideal distribution shape, suggesting possible prediction bias or inconsistency. Notable departures from normality were seen in the SOC predictions for Trips 23, 29, 39, 42, 55, and 62 and in battery voltage for Trips 13, 23, 29, 42, 55, and 57, also highlighted with red boxes in the figure to emphasize the increased variability and potential inconsistency in the model’s predictions.

These findings further reinforce the robustness of the TS-MSDA U-Net model. Its ability to maintain low error magnitudes and consistent distribution shapes across diverse driving scenarios makes it a more accurate and reliable solution for synthesizing key time series features in electric vehicle applications.

### 3.2. Reconstruction of Periodic Signals for Resonant CLLC Half-Bridge Converters

The resonant CLLC half-bridge converter is a highly efficient DC-DC topology widely adopted in applications requiring galvanic isolation and high-frequency operation. These applications include vehicle-to-grid (V2G), vehicle-to-home (V2H) systems, EV fast-charging infrastructure, and e-bike chargers.

As shown in Figure 6, the converter [27] employs an STM32F407VG digital signal processor (DSP) to implement soft-switching control based on both voltage and current feedback. A key architectural highlight is the symmetrical resonant tank, consisting of two inductors ($L_{1}$ and $L_{2}$) and two capacitors ($C_{1}$ and $C_{2}$) arranged on either side of a high-frequency transformer. This configuration enables efficient bidirectional power transfer while maintaining excellent voltage regulation across a wide range of operating conditions. The operation of this converter begins with a half-bridge inverter on the primary side, where two Gallium Nitride (GaN) switches ($S_{1}$ and $S_{2}$) alternate to produce a high-frequency square waveform. When $S_{1}$ is turned on and $S_{2}$ is off, energy is transferred from the input source through the primary inductor ($L_{1}$) and capacitor ($C_{1}$) into the transformer’s primary winding, initiating resonant energy transfer. During this phase, energy is temporarily stored in the resonant elements and delivered to the load via the transformer and a mirrored CLLC network on the secondary side. As the resonant current oscillates, a brief dead-time occurs—during which both switches are off—allowing the current to naturally reverse direction. This facilitates zero-voltage switching (ZVS) for the next switching cycle, significantly improving conversion efficiency. The converter operates near its resonant frequency, where impedance is minimized and energy transfer is most efficient.

The prototype CLLC converter operates at a switching frequency ($f_{pwm}$) of 250 kHz. Data acquisition is conducted using an oscilloscope at 500 MS/s via USB, with a resampled rate of 1.25 MHz Four channels are monitored: the drain-source voltage ($V_{ds}$) of the upper half-bridge switch on the primary side, the secondary-side capacitor voltage ($V_{c2}$), the output current ($I_{out}$) on the secondary side, and the primary-side capacitor voltage ($V_{c1}$). The primary capacitor voltage ($V_{c1}$) and the secondary-side capacitor voltage ($V_{c2}$) serve as critical indicators of the resonance characteristics of the LC tank circuits. However, a key limitation arises due to the insufficient number of ADC samples (only five) collected at 1.25 MHz, too few to capture the full behavior within a single PWM period. This limitation necessitates the use of signal reconstruction techniques to recover high-resolution periodic signals.

To address this challenge, the coprime sampling technique described in [28] is applied for high-resolution signal reconstruction. This method defines two coprime integers, N and M, based on the ratio between the sampling frequency ($f_{s}$) and the PWM frequency ($f_{pwm}$). The integer N represents the number of samples required to reconstruct one full period of the high-resolution signal, while M denotes the number of different PWM cycles across which these samples are collected. The reconstruction process arranges N samples collected across M PWM cycles into a full signal period using multiplicative inverse modulo, enabling high-fidelity waveform reconstruction from sparse samples.

The proposed TS-MSDA U-Net model offers several advantages over traditional coprime sampling techniques in real-world applications. It enables high-resolution reconstruction using only the DSP’s internal ADC, without the need for high-speed external samplers. Moreover, the model introduces operational flexibility by relaxing the requirement for strictly coprime $f_{s}$ and $f_{pwm}$, allowing deployment under more practical and varied sampling conditions. This flexibility enhances the model’s robustness to domain shift, as it can adapt to different hardware configurations and real-world signal variations.

Although this study primarily relied on simulation data, the model was developed with future deployment in mind. Subsequent work will include systematic testing on purely real hardware datasets and exploring domain adaptation techniques (e.g., fine-tuning or transfer learning) to explicitly quantify and address any simulation-to-reality performance gaps.

#### 3.2.1. Generation of Training Time Series Data Using the PLECS Simulator

PLECS (Piecewise Linear Electrical Circuit Simulation) [29] is a powerful simulation platform specifically tailored for modeling power electronic systems. Its ability to accurately replicate the switching behavior of semiconductor devices makes it particularly well-suited for simulating resonant converters such as the resonant CLLC half-bridge topology.

To obtain the time series data required for training the TS-MSDA U-Net model, a detailed simulation of the resonant CLLC half-bridge converter is constructed within the PLECS environment. This converter incorporates appropriately positioned measurement blocks to capture key electrical signals, including $V_{ds},$ $V_{C2}$, $I_{OUT},$ and $V_{C1}$. Time series data are extracted using the scope block’s export function available in PLECS. Once the simulation is executed across various load conditions and over a defined time interval, the generated data are saved as a CSV file for downstream processing.

Prior to model training, the raw simulation data undergo a preprocessing phase. This includes normalization to ensure consistent data ranges, and segmentation of the continuous waveforms into paired input-output sequences of 512 consecutive time steps, specifically tailored for the A/D resolution enhancement task. This structured data preparation ensures compatibility with the TS-MSDA U-Net training pipeline and enables effective learning of the underlying temporal patterns.

Ultimately, the simulation results from PLECS serve a dual purpose: They not only provide high-fidelity training data for the proposed model but also help validate the functional performance of the resonant CLLC half-bridge converter across diverse operating scenarios. These insights can later be corroborated through experimental validation using a physical prototype of the converter.

#### 3.2.2. Analysis of Training Experimental Results

Figure 7 illustrates a comparative analysis of the training and validation loss curves for five models developed to enhance the resolution of periodic signals in the resonant CLLC half-bridge converter. The plots use twin y-axes: the right axis features a logarithmic scale, allowing for a clearer representation of loss variations across a wide dynamic range.

Among the models, the baseline U-Net (red line) represents a standard convolutional architecture without specialized attention. U-Net with SA (blue line) incorporates an SA mechanism into the U-Net backbone. UNETR (green line) and UNETR++ (yellow line) are transformer-based models designed to capture long-range dependencies, with UNETR++ representing an improved variant. Finally, the TS-MSDA U-Net (purple line) integrates multiscale feature extraction and a dual-attention mechanism.

UNETR displays a unique loss pattern during the initial training phase. It experiences a sharp decline in training loss during the first few epochs, followed by a temporary increase before resuming a downward trend. This behavior suggests that the model rapidly captures basic data features but initially overfits to specific patterns, leading to transient instability before settling into more generalized learning.

A closer examination of the curves reveals that the baseline U-Net and UNETR maintain relatively high and stable loss values throughout training and validation. This indicates a limited ability to effectively learn and generalize from the training data. In contrast, the other models, which incorporate various advanced architectural elements, exhibit substantial performance improvements, with both training and validation losses consistently decreasing overtime. In contrast, the TS-MSDA U-Net, UNETR++, and U-Net with SA exhibit consistent declines in both training and validation losses, underscoring the effectiveness of their enhanced architectures. TS-MSDA U-Net and UNETR++ achieve the lowest final losses, especially evident in the later epochs when loss curves converge tightly. The use of a logarithmic scale on the right y-axis in Figure 7 further emphasizes their fine-grained performance improvements.

During validation, transient spikes were observed in some models (e.g., U-Net with SA and UNETR++), likely due to sensitivity to sequence variations. However, these models typically recover in subsequent epochs, suggesting resilience to temporary overfitting. Among all, TS-MSDA U-Net consistently achieved the lowest and most stable validation losses, demonstrating feature learning and predictive accuracy.

While these results highlight the benefits of advanced architectural integration, we acknowledge that this comparative analysis is not a full ablation study. A dedicated investigation to isolate the individual impact of components such as multiscale extraction, dual-attention, and shared projections remains an important avenue for future research.

Figure 8 illustrates the TS-MSDA U-Net model’s effectiveness in enhancing periodic signals under both heavy and light load conditions. The model synthesizes high-resolution signals from low-resolution input data generated through PLECS simulations, increasing the sampling density from 5 points per PWM period to 30 and 180 points per PWM period. These inputs are based on a PWM switching frequency of 250 kHz and a simulated ADC sampling frequency of 1.25 MHz. The leftmost panels of Figure 8 show the original low-resolution signals for four critical input channels: $V_{ds},$ $V_{C2}$, $I_{OUT},$ and $V_{C1}$. Each signal is plotted as a solid-colored line, with legend labels indicating the signal’s vertical scale (e.g., “$V_{ds}$ (100V)” denotes a grid scale of 100 V per vertical grid). The middle and rightmost panels display the corresponding synthetic signals generated by the TS-MSDA U-Net, overlaid with the ground truth data. In these enhanced panels, the model’s outputs are shown as solid-colored lines, while the ground truth waveforms are rendered as black dotted lines for direct visual comparison.

The PLECS simulations were conducted under 13 distinct loading conditions, with output current values ranging from 31 A (heavy load) to 1 A (light load). Under heavy load conditions, the synthetic signals reveal pronounced oscillations and dynamic behaviors across all channels. The output current ($I_{OUT}$) reaches approximately 31 A, and the resonant capacitor voltage ($V_{C1}$) swings dramatically within a range of [−200 V, 300 V]. In contrast, under light load conditions, ($V_{C1}$) is significantly reduced, fluctuating within a narrower range of [29 V, 72 V], and the waveforms exhibit more stable, less oscillatory behavior, with ($I_{OUT}$) near 1 A.

Despite the broad variation in operating conditions, the TS-MSDA U-Net demonstrates accuracy in reconstructing both coarse features and fine-grained high-frequency components of the signals. The model successfully recovers high-resolution characteristics from sparsely sampled inputs, underscoring its effectiveness in enhancing temporal resolution for precise signal analysis in the resonant CLLC half-bridge converters.

Figure 9 presents a comparative visualization of synthetic outputs produced by four distinct models: (a) U-Net, (b) UNETR, (c) U-Net with SA, and (d) UNETR++, evaluated against their corresponding ground truth signals. The results from models (a) and (b) clearly indicate limited reconstruction capability. Both the basic U-Net and UNETR models exhibit notable discrepancies and deviations in their synthetic outputs when compared to the ground true waveforms, reflecting an insufficient ability to capture the complex temporal dynamics inherent in the data. In contrast, the models enhanced with attention mechanisms—namely (c) U-Net with SA and (d) UNETR++—demonstrate markedly superior performance. Their reconstructed waveforms closely align with the ground truth signals, indicating that the incorporation of attention modules significantly improves the model’s ability to learn and replicate the underlying features of periodic signal data. This performance gain underscores the critical role of attention-based architectures in enhancing the fidelity of waveform reconstruction.

Furthermore, when compared to the TS-MSDA U-Net results presented earlier in Figure 8, both the U-Net with SA and UNETR++ models exhibit comparable and, in some instances, potentially superior reconstruction accuracy. Their ability to synthesize signals with high alignment to the ground truth highlights the effectiveness of attention mechanisms in modeling complex signal behavior. As a result, these models emerge as strong candidates for high-resolution time series reconstruction tasks in the applications requiring precise signal reconstruction under varying operating conditions.

A comprehensive evaluation of the five models—based on RMSE, MAE, and DTW metrics, as summarized in Table 4—highlights distinct differences in their performance when enhancing the resolution of periodic signals. These signals were obtained from simulations of a resonant CLLC half-bridge converter operating under thirteen diverse load conditions, ranging from heavy (Case 1) to light (Case 13).

Among the evaluated models, the basic U-Net and UNETR consistently exhibit the highest error values across all load scenarios, demonstrating limited ability to reconstruct accurate signals. Their elevated RMSE, MAE, and DTW values suggest insufficient capacity to capture the complex dynamics present in the time series data, thus resulting in lower fidelity reconstructions. In contrast, the models that incorporate attention mechanisms—namely, U-Net with SA, TS-MSDA U-Net, and UNETR++—show substantial improvements in reconstruction accuracy. These models achieve significantly lower error values across all metrics and load conditions. To further substantiate these performance differences, statistical tests were conducted using two-tailed paired t-tests across all 13 cases. The resulting *p*-values are provided in Table 5. The TS-MSDA U-Net shows statistically significant improvements over U-Net, U-Net with SA, and UNETR in terms of RMSE, MAE, and DTW, with all *p*-values falling well below the 0.05 threshold. This confirms that the performance gains observed are not merely due to random variability but reflect meaningful improvements in signal reconstruction fidelity. When compared to UNETR++, TS-MSDA U-Net demonstrates statistically significant improvement in MAE (*p* = 3.23 × 10⁻³), while the differences in RMSE (*p* = 0.422) and DTW (*p* = 0.242) are not statistically significant. This indicates that, although TS-MSDA U-Net generally performs better across all metrics, UNETR++ achieves comparable results in certain scenarios, particularly in global and alignment-based measures. The TS-MSDA U-Net’s superior performance can be attributed to its dual-attention mechanisms and multiscale feature extraction, which enable the model to effectively learn complex temporal dependencies and capture signal features across varying resolutions. While UNETR++ and U-Net with SA demonstrate better accuracy than its baseline variants, it generally falls short of the TS-MSDA U-Net’s performance. Nevertheless, it still achieves a marked improvement over the traditional approach.

Table 6 presents a per-signal breakdown of RMSE, MAE, and DTW metrics to evaluate the TS-MSDA U-Net’s reconstruction performance on the four key channels: $V_{ds},$ $V_{C2}$, $I_{OUT},$ and $V_{C1}$. The model demonstrates consistently low error values across all three metrics, indicating reconstruction accuracy. Notably, $V_{C2}$ and $V_{C1}$ exhibit the lowest RMSE and DTW values among the four signals, reflecting more stable amplitude estimation and temporal alignment. Although $I_{OUT}$ initially presents higher RMSE, its error decreases steadily in subsequent cases, suggesting the model’s capacity to adapt to more complex signal patterns. The overall downward trend in error metrics from Case 1 to Case 13 highlights the model’s generalization across varying signal dynamics and driving conditions.

#### 3.2.3. Testing Experimental Results Using the Prototype Converters

Figure 10 presents a visual comparison between the synthetic signals generated by the TS-MSDA U-Net and the corresponding ground truth measurements obtained from a prototype CLLC half-bridge converter operating under both heavy and light load conditions. The ground truth signals were captured using an oscilloscope with a sampling rate of 500 MS/s via USB, as shown in the upper-left panel of the figure. To ensure all relevant signals ($V_{ds}$, $V_{C1}$, and $V_{C2}$) were clearly displayed, the oscilloscope grid scales were adjusted according to the load conditions.

Under heavy load operation, the synthetic data generated by the TS-MSDA U-Net exhibit a high degree of fidelity, with waveform shapes that closely match those of the real measured signals. A minor discrepancy is observed in the $V_{ds}$ waveform, where the synthetic output displays sharper transitions in the square waveforms compared to the more rounded edges of the measured signals, likely due to parasitic effects not captured in the simulation-based training data. Under light load conditions, the differences between the synthetic and ground truth signals become more pronounced. The overall accuracy of waveform reconstruction is reduced, with a noticeable phase shift appearing in the $V_{ds}$ signal, indicating a slight temporal misalignment between the generated and actual data.

The input provided to the TS-MSDA U-Net was at a low resolution of only five points per PWM period, corresponding to an ADC sampling rate of 1.25 MHz. Despite this sparse input, the model successfully reconstructs high-resolution waveforms with 30 and 180 points per period, equivalent to effective sampling rates of 7.5 MHz and 45 MHz, respectively. These results underscore the TS-MSDA U-Net’s capability to significantly enhance temporal resolution, even under real-world operating conditions.

Table 7 provides a comparative summary of model complexity and efficiency, including parameter counts, average training time per epoch, and per-sample inference speed across all evaluated models. While TS-MSDA U-Net has a higher parameter count and longer training time compared to simpler architectures such as U-Net and U-Net with SA, its inference speed (0.0349 s/sample) is significantly faster, second only to UNETR. The increase in parameter count is primarily due to the dual-attention modules and multi-branch architecture, which are instrumental in achieving performance, as demonstrated by the evaluation metrics.

## 4. Conclusions

The TS-MSDA U-Net is a novel architecture specifically designed to capture complex temporal dynamics in long-range TSS tasks. It combines a hierarchical encoder–decoder structure for multiscale temporal feature extraction with a dual-attention mechanism, incorporating both SA and CA to enhance the modeling capacity. The performances of the mode were rigorously benchmarked against several baseline architectures, including the basic U-Net, U-Net with SA, UNETR, and UNETR++, using RMSE, MAE, and DTW as key evaluation metrics.

In the first application, synthesizing multivariate time series data from 70 real-world EV driving trips, the TS-MSDA U-Net achieved demonstrated notable improvements. It achieved MAEs below 1% for key parameters such as SOC, battery voltage, mechanical acceleration, and motor torque, representing a two-fold improvement over the TS-p2pGAN baseline. While violin plot analyses revealed greater variability in TS-p2pGAN, particularly with deviations up to 5% in motor torque and longitudinal acceleration, the TS-MSDA U-Net showed significantly more stable accuracy. The U-Net with SA and UNETR models exhibited comparatively higher training losses, likely due to their sensitivity to temporal fluctuations or increased architectural complexity. Although UNETR++ occasionally matched the TS-MSDA U-Net in some metrics, the proposed model consistently outperformed the basic U-Net. It is worth noting, however, that the gains over the basic U-Net were relatively modest. This finding suggests that, for less complex or noise-tolerant tasks, the computational overhead introduced by the dual-attention and multiscale mechanisms may not be strictly necessary. The effectiveness of such enhancements is therefore task-dependent and should be evaluated in light of the application’s accuracy requirements and resource constraints.

In contrast, the second application, high-resolution signal reconstruction in a resonant CLLC half-bridge converter, unequivocally demonstrated the necessity of the TS-MSDA U-Net’s architectural complexity. The model successfully reconstructed high-frequency periodic waveforms from low-resolution ADC inputs, achieving a remarkable 36× improvement in effective temporal resolution. Competing architecture, including basic U-Net and UNETR, failed to produce accurate outputs. This success confirms that, for applications demanding fine-grained signal fidelity and robust temporal modeling, the increased parameter count and training cost of the TS-MSDA U-Net are fully justified.

Despite its advantages, the TS-MSDA U-Net has limitations. Its high parameter count and training time can challenge deployment in resource-constrained environments. Furthermore, its evaluation has been confined to two domain-specific tasks—EV time series synthesis and high-resolution signal reconstruction in power electronics—leaving its generalizability to other time series domains unverified. Additionally, the current framework relies on fully supervised learning, which may restrict its use in data-scarce scenarios.

In summary, the TS-MSDA U-Net offers a powerful solution for multivariate time series modeling in engineering systems with nonlinear and temporally complex behaviors. While its benefits may be marginal for simpler tasks, its superior performance in high-resolution reconstruction tasks underscores its value in precision-critical applications. Future work may explore pruning strategies or lightweight variants to balance accuracy and efficiency across diverse deployment scenarios.

## Figures and Tables

**Figure 1 sensors-25-04073-f001:**
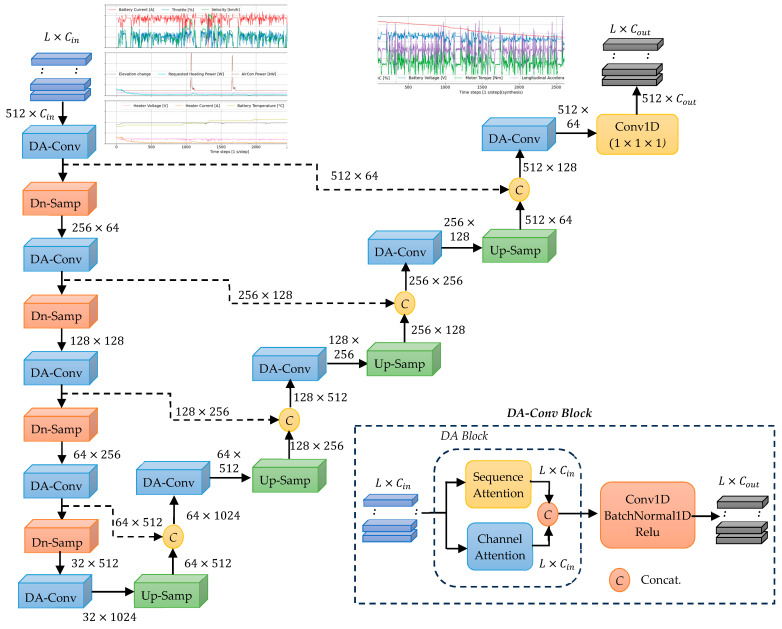
Overview of the TS-MSDA U-Net architecture.

**Figure 2 sensors-25-04073-f002:**
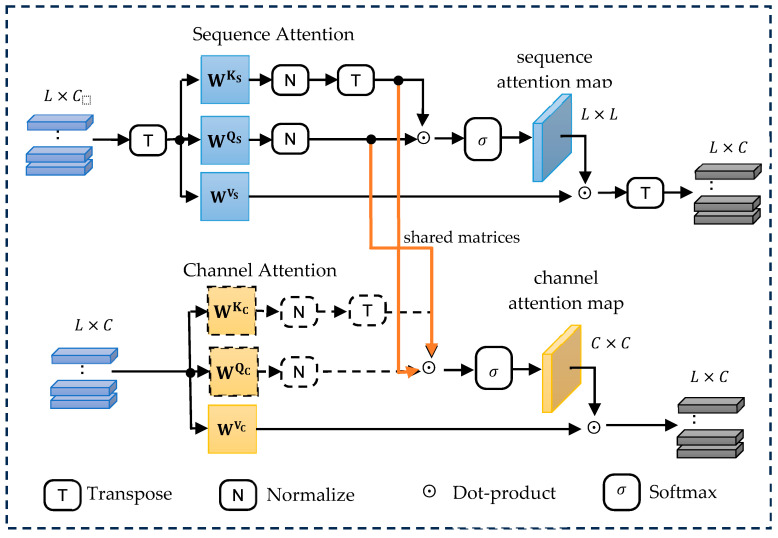
Architecture of the DA block.

**Figure 3 sensors-25-04073-f003:**
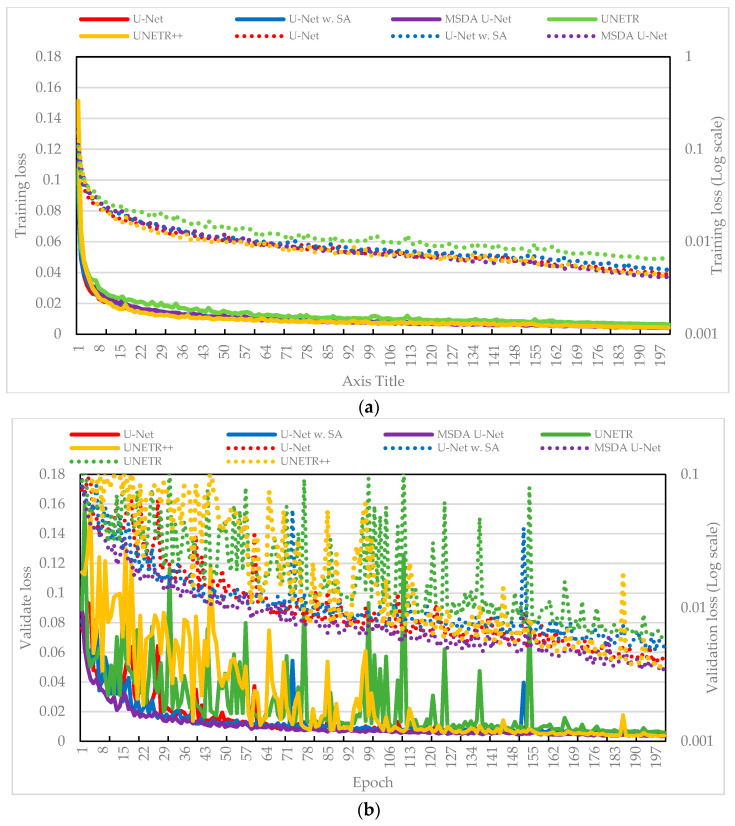
(**a**) Training and (**b**) validation loss curves obtained during the training phase. A logarithmic scale is used for the vertical axis to emphasize small variations in loss values across models, particularly during the later stages of training. This enhances the visibility of convergence behavior and fine-grained performance differences that may be obscured on a linear scale.

**Figure 4 sensors-25-04073-f004:**
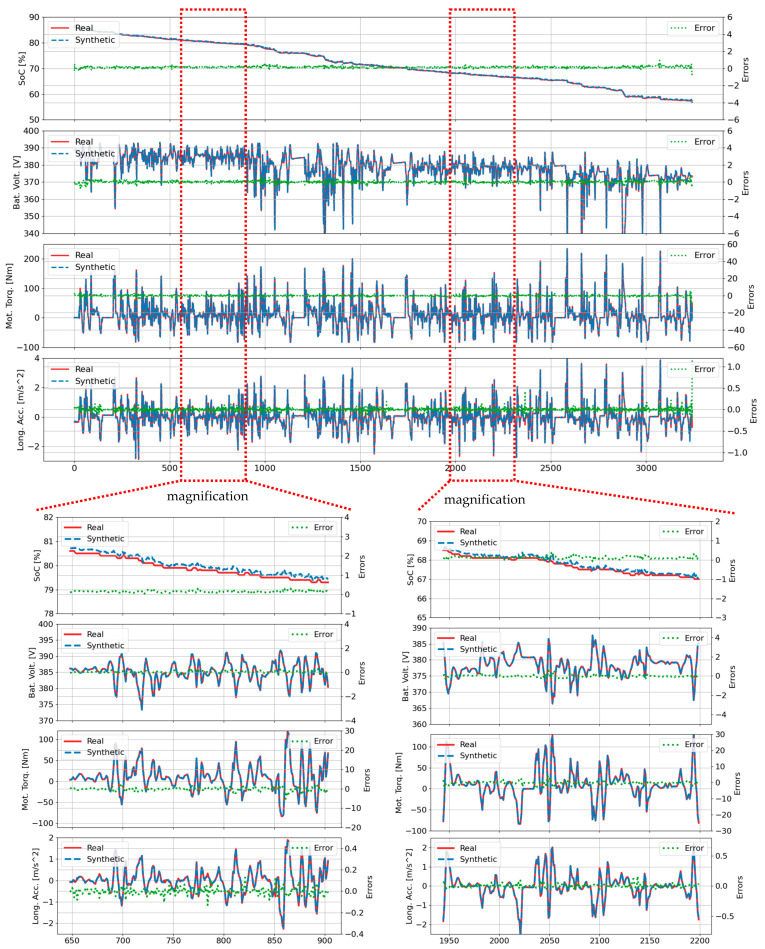
Real and synthetic time series data generated by the TS-MSDA U-Net model for the B01 trip. The error values shown are in absolute units consistent with the original signal scale (e.g., SOC in percentage points), not normalized or expressed as percentages. This allows for direct interpretation without distortion from trip-specific range variability.

**Figure 5 sensors-25-04073-f005:**
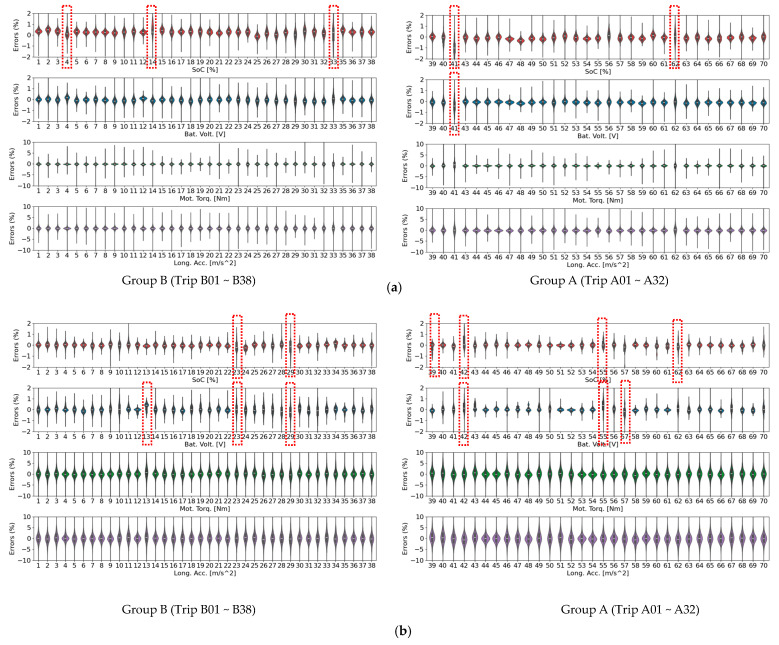
Violin plots comparing error distributions of four features across all trips for (**a**) TS-MSDA U-Net and (**b**) TS-p2pGAN. The red dashed boxes mark trips where the error distributions exhibit visible deviations from the typical bell-shaped, symmetrical pattern observed in most other cases. These deviations may indicate localized prediction inconsistencies or input-specific sensitivity.

**Figure 6 sensors-25-04073-f006:**
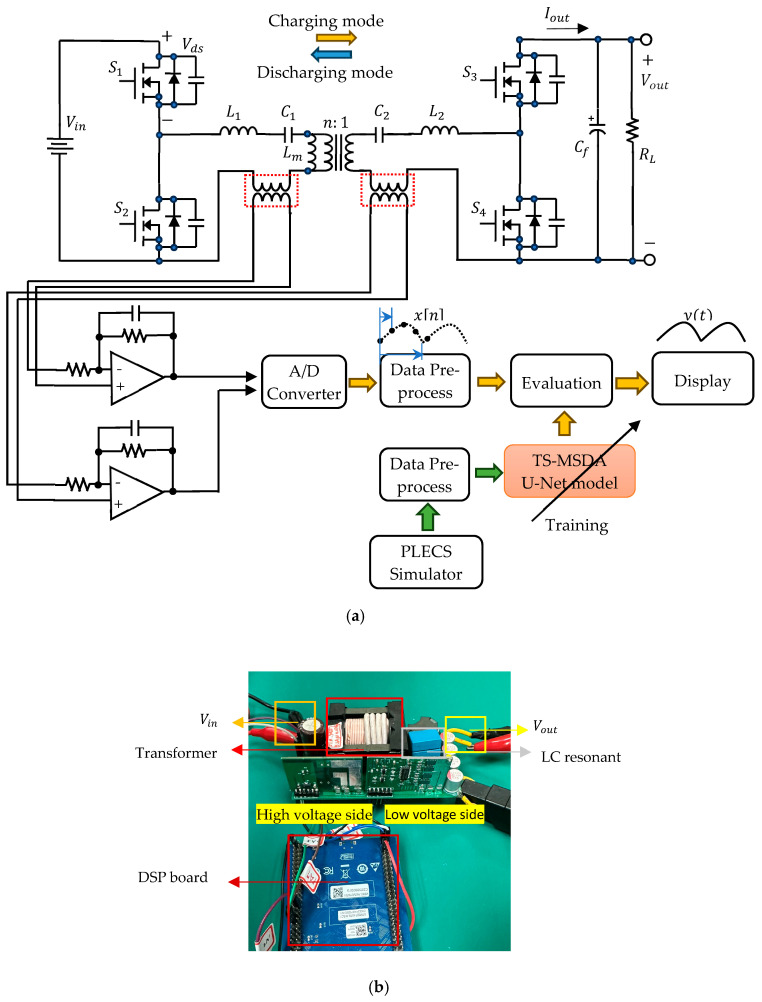
TS-MSDA U-Net architecture applied to enhance A/D resolution in a resonant CLLC half-bridge converter. (**a**) Circuit schematic. (**b**) Prototype photograph.

**Figure 7 sensors-25-04073-f007:**
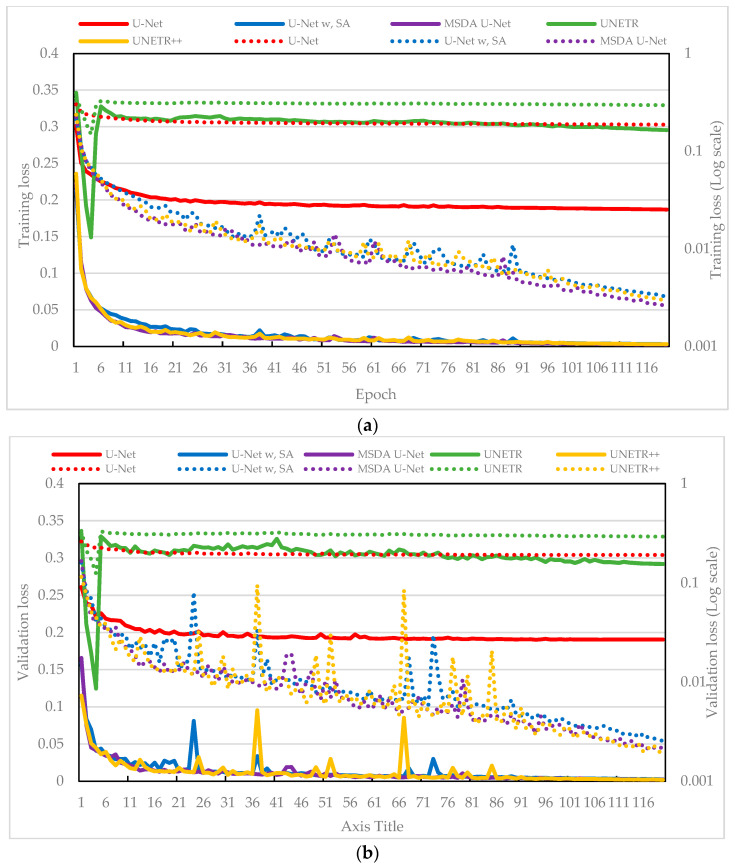
(**a**) Training and (**b**) validation loss curves for enhancing the resolution of periodic signals in resonant CLLC half-bridge converters.

**Figure 8 sensors-25-04073-f008:**
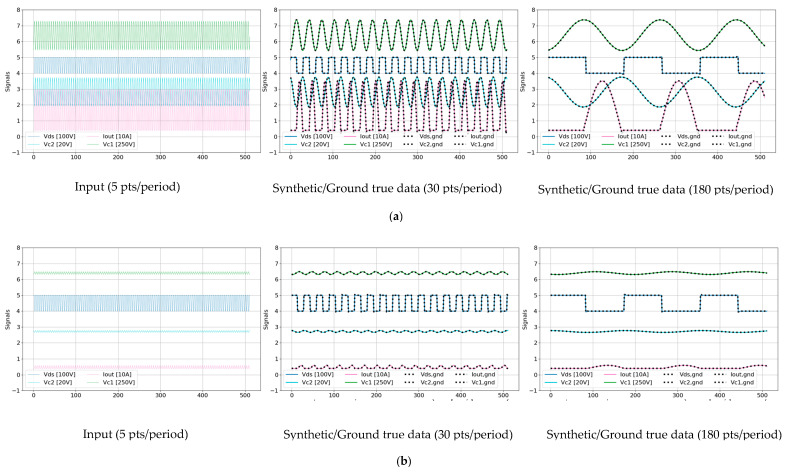
Enhancement of 512-point segments using the TS-MSDA U-Net. The model increases the sampling density from a low resolution of 5 points/period to 30 points/period and 180 points/period under (**a**) heavy load and (**b**) light load operating conditions.

**Figure 9 sensors-25-04073-f009:**
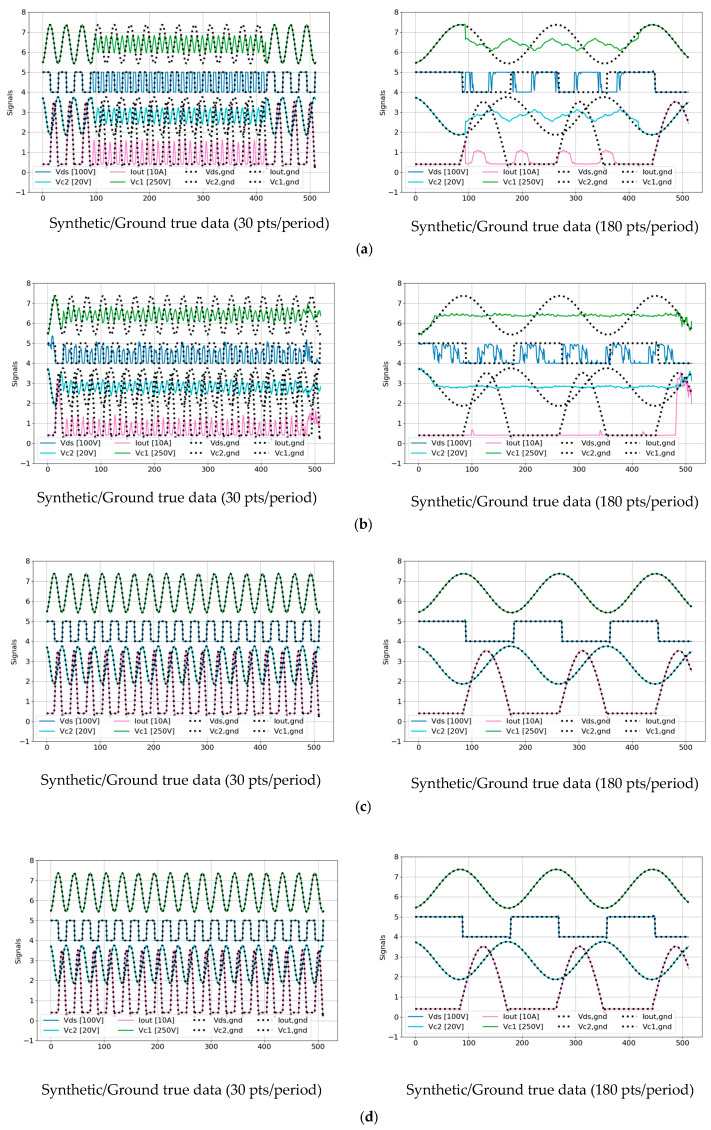
Visual comparisons of synthetic outputs against ground truth data across various models: (**a**) U-Net, (**b**) UNETR, (**c**) U-Net with SA, and (**d**) UNETR++.

**Figure 10 sensors-25-04073-f010:**
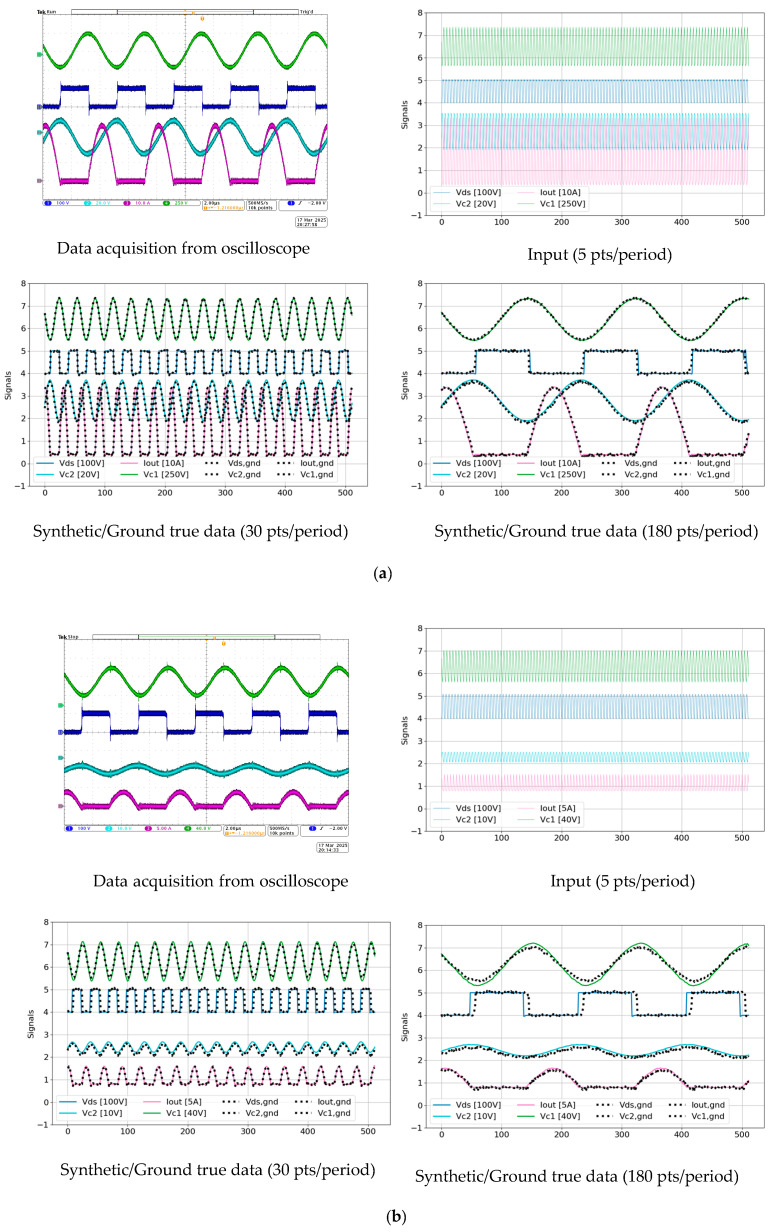
Visual assessment of TS-MSDA U-Net synthetic data against ground truth in a prototype CLLC converter under various load conditions: (**a**) heavy load and (**b**) light load.

**Table 1 sensors-25-04073-t001:** Layer-wise hyperparameters of the TS-MSDA U-Net.

Name	Layer	(k, s, p)	L	C	Module
Input			512	$C_{in}$	Encoder section
DA-Conv_1	DA block	(-, -, -)	512	$2\cdot C_{in}$	
	Conv1d_N+R_	(3, 1, 1)	512	64	
Dn-Sample_1	MaxPool1D	(2, 2, 1)	256	64	
DA-Conv-2	DA block	(-, -, -)	256	128	
	Conv1d_N+R_	(3, 1, 1)	256	128	
Dn-Sample-2	MaxPool1D	(2, 2, 1)	128	128	
DA-Conv-3	DA block	(-, -, -)	128	256	
	Conv1d_N+R_	(3, 1, 1)	128	256	
Dn-Sample-3	MaxPool1D	(2, 2, 1)	64	512	
DA-Conv-4	DA block	(-, -, -)	64	512	
	Conv1d_N+R_	(3, 1, 1)	64	512	
Dn-Sample-4	MaxPool1D	(2, 2, 1)	32	512	
DA-Conv-5	DA block	(-, -, -)	32	1024	Bottleneck
	Conv1d_N+R_	(3, 1, 1)	32	1024	
Up-Sample-1	ConvTranspose1d	(2, 2, 1)	64	512	Decoder section
DA-Conv-5	DA block	(-, -, -)	64	1024	
	Conv1d_N+R_	(3, 1, 1)	64	512	
Up-Sample-2	ConvTranspose1d	(2, 2, 1)	128	256	
DA-Conv-6	DA block	(-, -, -)	128	512	
	Conv1d_N+R_	(3, 1, 1)	128	256	
Up-Sample-3	ConvTranspose1d	(2, 2, 1)	256	128	
DA-Conv-7	DA block	(-, -, -)	256	256	
	Conv1d_N+R_	(3, 1, 1)	256	128	
Up-Sample-4	ConvTranspose1d	(2, 2, 1)	512	64	
DA-Conv-8	DA block	(-, -, -)	512	128	
	Conv1d_N+R_	(3, 1, 1)	512	64	
Output	Conv1D	(1,1,1)	512	$C_{out}$	Output Layer

Note: k: kernel, s; stride, p: padding, f: feature, d: dimension. Conv1dN + R: Conv1d + BatchNorm1d + ReLU.

**Table 2 sensors-25-04073-t002:** RMSE, MAE, and DTW values for all trips.

Trip No	U-Net			U-Net with SA			TS-MSDA-U-Net			UNETR			UNETR++			TS-p2pGAN		
	RMSE	MAE	DTW	RMSE	MAE	DTW	RMSE	MAE	DTW	RMSE	MAE	DTW	RMSE	MAE	DTW	RMSE	MAE	DTW
1	0.96	0.45	0.54	1.05	0.52	0.63	0.68	0.39	0.47	1.25	0.61	0.76	0.75	0.37	0.47	1.96	0.97	1.19
2	0.71	0.46	0.53	0.80	0.49	0.57	0.65	0.42	0.51	1.10	0.63	0.69	0.68	0.36	0.46	2.02	1.01	1.12
3	0.72	0.45	0.53	0.83	0.47	0.58	0.59	0.39	0.47	1.17	0.60	0.73	0.65	0.38	0.47	1.91	1.02	1.21
4	1.07	0.42	0.48	1.21	0.47	0.53	0.66	0.35	0.36	1.40	0.51	0.58	0.75	0.35	0.39	1.79	0.77	0.84
5	1.34	0.54	0.65	1.43	0.60	0.74	0.92	0.45	0.55	1.72	0.67	0.83	1.09	0.46	0.59	1.91	0.92	1.09
6	0.81	0.46	0.55	1.01	0.49	0.61	0.65	0.38	0.47	1.11	0.57	0.70	0.78	0.39	0.51	1.62	0.84	1.03
7	0.76	0.42	0.50	0.83	0.44	0.54	0.67	0.39	0.46	1.00	0.54	0.65	0.71	0.38	0.48	1.56	0.84	1.01
8	0.74	0.40	0.47	0.81	0.41	0.50	0.60	0.34	0.41	0.93	0.47	0.56	0.64	0.32	0.40	1.51	0.78	0.92
9	0.77	0.43	0.48	0.86	0.44	0.50	0.54	0.32	0.37	1.08	0.48	0.55	0.62	0.32	0.39	1.73	0.85	0.95
10	1.02	0.51	0.62	1.19	0.53	0.67	0.74	0.41	0.51	1.40	0.63	0.78	0.92	0.42	0.54	2.03	1.06	1.26
11	1.14	0.54	0.61	1.14	0.46	0.55	0.81	0.42	0.49	1.69	0.65	0.74	0.86	0.39	0.47	2.08	1.01	1.15
12	0.72	0.36	0.43	0.78	0.39	0.46	0.56	0.35	0.40	0.96	0.44	0.51	0.62	0.32	0.39	1.26	0.66	0.72
13	---	---	---	---	---	---	---	---	---	---	---	---	---	---	---	2.69	1.46	1.56
14	1.28	0.44	0.51	1.36	0.45	0.53	1.19	0.39	0.46	1.54	0.55	0.64	1.08	0.34	0.43	1.48	0.80	0.91
15	0.90	0.47	0.56	1.15	0.50	0.60	0.67	0.42	0.50	1.31	0.62	0.76	0.66	0.38	0.49	1.83	0.96	1.17
16	0.73	0.43	0.52	0.82	0.46	0.57	0.62	0.39	0.48	1.04	0.59	0.74	0.69	0.38	0.49	1.92	1.02	1.21
17	0.79	0.48	0.59	0.83	0.44	0.57	0.64	0.41	0.50	1.06	0.58	0.74	0.67	0.39	0.50	2.11	1.07	1.28
18	1.16	0.53	0.63	1.26	0.52	0.63	0.68	0.43	0.52	1.50	0.63	0.73	0.78	0.41	0.52	1.66	0.87	1.01
19	0.84	0.47	0.56	0.97	0.52	0.62	0.66	0.39	0.48	1.24	0.60	0.72	0.71	0.38	0.48	1.80	0.92	1.09
20	0.76	0.43	0.51	0.94	0.49	0.59	0.58	0.37	0.45	1.25	0.62	0.76	0.80	0.38	0.48	1.91	0.96	1.11
21	0.97	0.45	0.55	1.20	0.46	0.57	0.68	0.36	0.44	1.51	0.58	0.71	0.83	0.37	0.47	1.56	0.80	0.97
22	1.00	0.47	0.57	1.15	0.49	0.61	0.68	0.39	0.49	1.25	0.60	0.73	0.82	0.41	0.53	2.05	0.97	1.20
23	1.07	0.54	0.66	1.37	0.59	0.74	0.86	0.46	0.57	1.76	0.73	0.90	1.09	0.49	0.63	2.65	1.31	1.52
24	1.46	0.59	0.73	1.98	0.67	0.88	0.81	0.48	0.60	2.34	0.90	1.14	1.07	0.50	0.66	2.36	1.25	1.42
25	0.78	0.49	0.60	0.80	0.45	0.56	0.60	0.37	0.46	1.03	0.56	0.68	0.66	0.37	0.48	2.23	1.09	1.26
26	1.53	0.64	0.76	1.56	0.64	0.77	1.09	0.46	0.56	2.55	0.90	1.07	1.22	0.46	0.60	2.75	1.30	1.51
27	1.22	0.59	0.70	1.47	0.60	0.72	0.89	0.43	0.52	1.68	0.71	0.86	0.89	0.48	0.58	2.04	1.11	1.30
28	0.96	0.64	0.74	0.98	0.62	0.72	0.70	0.43	0.53	1.30	0.65	0.79	0.75	0.42	0.54	2.21	1.23	1.40
29	0.92	0.59	0.68	1.04	0.61	0.70	0.69	0.45	0.53	1.20	0.67	0.79	0.77	0.51	0.60	2.90	1.67	1.77
30	1.54	0.65	0.76	1.89	0.66	0.77	0.87	0.49	0.58	1.85	0.75	0.89	0.97	0.54	0.64	2.25	1.02	1.22
31	1.15	0.54	0.67	1.29	0.56	0.70	0.64	0.42	0.51	1.54	0.65	0.82	0.74	0.41	0.54	1.86	0.87	1.06
32	1.15	0.56	0.67	1.49	0.60	0.75	0.92	0.51	0.61	1.90	0.73	0.88	0.99	0.48	0.60	2.70	1.30	1.52
33	1.89	0.89	1.04	2.34	0.91	1.08	1.52	0.76	0.91	2.15	0.99	1.17	1.47	0.74	0.91	2.45	1.24	1.60
34	---	---	---	---	---	---	---	---	---	---	---	---	---	---	---	2.32	1.12	1.32
35	1.02	0.51	0.62	1.15	0.56	0.67	0.71	0.44	0.54	1.46	0.64	0.79	0.79	0.40	0.53	1.95	0.93	1.08
36	0.90	0.43	0.52	1.08	0.45	0.56	0.71	0.37	0.46	1.26	0.55	0.67	0.84	0.36	0.46	1.75	0.86	1.00
37	1.19	0.53	0.62	1.44	0.60	0.71	0.79	0.42	0.52	1.84	0.73	0.90	0.87	0.43	0.53	1.89	0.93	1.13
38	0.94	0.51	0.63	1.14	0.55	0.69	0.70	0.41	0.50	1.52	0.66	0.83	0.81	0.43	0.55	2.30	1.13	1.36
39	1.04	0.53	0.66	1.21	0.57	0.70	0.80	0.41	0.52	1.28	0.65	0.80	0.87	0.44	0.57	2.17	1.18	1.40
40	1.36	0.81	0.95	1.78	1.10	1.35	1.05	0.60	0.72	2.22	1.40	1.66	0.97	0.57	0.71	2.45	1.33	1.67
41	2.42	1.26	1.41	4.71	2.10	2.34	2.81	1.42	1.52	5.57	2.94	3.34	2.45	1.03	1.17	2.40	1.19	1.48
42	---	---	---	---	---	---	---	---	---	---	---	---	---	---	---	2.71	1.26	1.40
43	0.87	0.45	0.58	1.12	0.58	0.73	0.76	0.39	0.51	1.40	0.68	0.84	0.91	0.45	0.60	2.33	1.22	1.48
44	0.81	0.37	0.47	1.01	0.49	0.59	0.60	0.32	0.40	1.19	0.50	0.63	0.72	0.34	0.45	1.47	0.77	0.90
45	0.60	0.38	0.47	0.74	0.46	0.56	0.53	0.33	0.41	0.86	0.52	0.64	0.61	0.35	0.45	1.58	0.79	0.93
46	0.89	0.43	0.52	1.13	0.58	0.66	0.62	0.32	0.40	1.13	0.54	0.66	0.70	0.37	0.47	2.41	1.20	1.39
47	1.06	0.46	0.57	1.58	0.62	0.78	0.72	0.36	0.45	1.78	0.65	0.84	0.87	0.42	0.56	2.28	1.08	1.30
48	1.21	0.49	0.63	1.40	0.60	0.75	0.90	0.44	0.54	1.57	0.67	0.84	1.05	0.47	0.61	1.72	0.85	1.00
49	0.76	0.40	0.48	0.93	0.49	0.58	0.58	0.34	0.41	0.98	0.51	0.63	0.70	0.36	0.46	2.58	1.24	1.45
50	1.14	0.49	0.62	1.65	0.63	0.80	0.72	0.38	0.47	2.14	0.78	0.98	0.95	0.46	0.60	3.29	1.46	1.75
51	1.24	0.57	0.73	1.60	0.67	0.86	0.92	0.47	0.60	2.47	0.79	1.01	1.12	0.52	0.68	2.19	1.15	1.39
52	1.13	0.55	0.68	1.37	0.68	0.83	0.79	0.44	0.55	1.52	0.73	0.92	0.95	0.52	0.67	2.21	1.09	1.30
53	0.67	0.37	0.47	0.85	0.49	0.60	0.58	0.33	0.42	1.04	0.52	0.66	0.69	0.36	0.47	1.58	0.79	0.94
54	0.86	0.40	0.48	1.05	0.51	0.60	0.55	0.33	0.40	1.39	0.56	0.66	0.84	0.35	0.44	1.45	0.73	0.87
55	0.56	0.38	0.45	0.61	0.41	0.49	0.54	0.35	0.43	0.65	0.44	0.52	0.52	0.31	0.40	2.31	1.14	1.27
56	1.01	0.48	0.58	1.32	0.61	0.73	0.75	0.45	0.55	1.41	0.71	0.84	0.73	0.45	0.56	1.48	0.84	0.99
57	0.99	0.40	0.50	1.22	0.54	0.64	0.74	0.34	0.42	1.45	0.58	0.72	0.72	0.35	0.45	2.54	1.24	1.45
58	0.88	0.40	0.51	1.55	0.54	0.67	0.56	0.32	0.41	1.91	0.60	0.76	0.71	0.38	0.49	1.75	0.94	1.15
59	0.81	0.46	0.57	1.01	0.56	0.69	0.76	0.42	0.53	1.02	0.57	0.71	0.80	0.44	0.57	1.55	0.79	0.97
60	0.99	0.45	0.55	1.24	0.55	0.66	0.76	0.36	0.45	1.54	0.63	0.77	1.14	0.46	0.57	2.05	1.00	1.19
61	0.83	0.39	0.49	1.21	0.54	0.65	0.80	0.36	0.45	1.57	0.57	0.71	0.87	0.37	0.50	2.05	1.03	1.27
62	1.72	0.91	1.04	1.84	1.09	1.24	1.84	0.85	1.01	1.80	0.95	1.12	1.45	0.76	0.94	2.78	1.27	1.43
63	0.88	0.55	0.65	0.99	0.70	0.80	0.65	0.40	0.50	1.07	0.65	0.80	0.78	0.42	0.55	1.87	0.99	1.21
64	0.84	0.47	0.58	0.99	0.56	0.67	0.64	0.37	0.46	1.15	0.62	0.77	0.75	0.40	0.53	2.04	1.06	1.35
65	0.86	0.46	0.58	1.05	0.59	0.71	0.61	0.35	0.45	1.38	0.62	0.80	0.75	0.40	0.53	2.52	1.16	1.45
66	1.28	0.52	0.66	1.44	0.66	0.80	0.74	0.40	0.51	1.79	0.67	0.85	0.93	0.44	0.58	1.63	0.87	1.09
67	0.77	0.44	0.55	0.93	0.51	0.62	0.57	0.34	0.43	1.49	0.58	0.72	0.92	0.39	0.50	2.74	1.37	1.69
68	1.13	0.55	0.67	1.40	0.64	0.79	0.81	0.41	0.53	1.69	0.74	0.93	0.94	0.47	0.62	2.16	1.10	1.35
69	0.81	0.47	0.59	0.98	0.65	0.76	0.65	0.38	0.49	1.09	0.58	0.75	0.79	0.42	0.56	1.88	0.96	1.22
70	1.07	0.51	0.62	1.25	0.60	0.73	0.82	0.38	0.48	2.02	0.73	0.91	0.85	0.41	0.54	2.37	1.13	1.34

Note: The data generated by TS-p2pGAN, sourced from reference [7], consist of sequences with a length of 256. In this study, the sequence length is extended to 512. Trips 13, 34, and 42 are excluded due to their insufficient duration. Group B trips, labeled B01–B36, correspond to trip numbers 1–36, while Group A trips, labeled A01–A32, correspond to trip numbers 37–70.

**Table 3 sensors-25-04073-t003:** The *p*-values for the pairwise comparisons of RMSE, MAE, and DTW between TS-MSDA U-Net and other models across 70 trips. Statistical significance is assessed via two-tailed paired t-tests.

Model	RMSE	MAE	DTW	Interpretation
U-Net	$1.06\times 10^{-18}$	$1.74\times 10^{-20}$	$9.68\times 10^{-23}$	Significant difference (*p* < 0.05)
U-Net with SA	$1.13\times 10^{-20}$	$1.20\times 10^{-20}$	$1.08\times 10^{-20}$	Significant difference (*p* < 0.05)
UNETR	$1.54\times 10^{-22}$	$2.39\times 10^{-17}$	$1.43\times 10^{-17}$	Significant difference (*p* < 0.05)
UNETR++	$8.22\times 10^{-8}$	$4.48\times 10^{-1}$	$2.18\times 10^{-4}$	Mixed results; MAE not significant
Ts-p2pGAN	$8.04\times 10^{-34}$	$4.74\times 10^{-34}$	$7.99\times 10^{-36}$	Significant difference (*p* < 0.05)

**Table 4 sensors-25-04073-t004:** RMSE, MAE, and DTW metrics for evaluating resolution enhancement in bidirectional half-bridge CLLC converters.

Case No	U-Net			U-Net with SA			TS-MSDA-UNet			UNETR			UNETR++		
	RMSE (%)	MAE (%)	DTW (%)	RMSE (%)	MAE (%)	DTW (%)	RMSE (%)	MAE (%)	DTW (%)	RMSE (%)	MAE (%)	DTW (%)	RMSE (%)	MAE (%)	DTW (%)
1	52.60	30.19	20.17	1.34	0.44	0.47	0.84	0.40	0.41	63.37	45.57	31.28	1.05	0.43	0.47
2	49.11	28.04	18.80	1.15	0.31	0.34	0.90	0.33	0.35	59.09	42.35	29.26	0.89	0.37	0.37
3	46.20	26.28	17.59	1.36	0.29	0.33	0.82	0.30	0.32	55.48	39.57	27.30	1.04	0.33	0.33
4	43.30	24.49	16.09	1.41	0.29	0.32	0.67	0.26	0.28	52.14	37.12	25.38	0.65	0.28	0.29
5	39.79	22.32	14.85	1.35	0.26	0.28	0.65	0.24	0.25	48.45	34.49	23.23	0.48	0.24	0.24
6	37.20	20.60	13.65	1.24	0.24	0.27	0.81	0.22	0.24	45.25	31.98	21.45	0.69	0.22	0.22
7	34.74	18.90	12.41	1.30	0.24	0.27	0.72	0.20	0.22	42.21	29.49	19.63	0.74	0.20	0.20
8	31.68	16.60	10.70	1.36	0.25	0.29	0.53	0.18	0.19	38.54	26.28	17.18	0.51	0.18	0.18
9	29.62	14.86	9.40	1.48	0.23	0.27	0.54	0.17	0.18	35.65	23.49	15.33	0.63	0.17	0.17
10	27.76	13.10	8.11	1.07	0.22	0.25	0.55	0.16	0.18	33.62	21.14	13.49	0.68	0.18	0.18
11	26.24	11.34	6.85	0.51	0.18	0.19	0.30	0.14	0.14	32.09	18.74	11.65	0.28	0.16	0.15
12	25.12	9.56	5.56	0.39	0.16	0.16	0.27	0.12	0.12	30.92	16.18	9.80	0.27	0.15	0.14
13	24.47	7.89	4.32	0.41	0.13	0.14	0.25	0.10	0.10	30.16	13.62	8.09	0.28	0.16	0.15

Note: Case 1 represents a heavy load condition, while Case 13 corresponds to a light load condition. The sequence of cases transitions progressively from heavy load to light load.

**Table 5 sensors-25-04073-t005:** The *p*-values for the pairwise comparisons of RMSE, MAE, and DTW between TS-MSDA U-Net and other models across 13 cases. Statistical significance is assessed via two-tailed paired *t*-tests.

Model	RMSE	MAE	DTW	Interpretation
U-Net	$1.16\times 10^{-8}$	$7.51\times 10^{-7}$	$2.08\times 10^{-6}$	Significant difference (*p* < 0.05)
U-Net with SA	$1.61\times 10^{-5}$	$7.23\times 10^{-4}$	$1.22\times 10^{-4}$	Significant difference (*p* < 0.05)
UNETR	$8.87\times 10^{-9}$	$2.78\times 10^{-7}$	$8.31\times 10^{-7}$	Significant difference (*p* < 0.05)
UNETR++	$4.22\times 10^{-1}$	$3.23\times 10^{-3}$	$2.42\times 10^{-1}$	Mixed results; RMSE and DTW not significant

**Table 6 sensors-25-04073-t006:** Evaluation of the four-channel signals: $V_{ds},$ $V_{C2}$, $I_{OUT,}$ and $V_{C1}$ using RMSE, MAE, and DTW metrics achieved by the TS-MSDA U-Net model.

Case No	$V_{ds}$			$V_{C2}$			$I_{OUT}$			$V_{C1}$		
	RMSE (%)	MAE (%)	DTW (%)	RMSE (%)	MAE (%)	DTW (%)	RMSE (%)	MAE (%)	DTW (%)	RMSE (%)	MAE (%)	DTW (%)
1	0.71	0.17	0.20	0.54	0.38	0.60	1.27	0.62	0.95	0.61	0.43	0.68
2	1.41	0.16	0.22	0.43	0.31	0.49	0.93	0.51	0.80	0.48	0.34	0.54
3	1.32	0.15	0.22	0.37	0.27	0.42	0.81	0.46	0.72	0.41	0.30	0.48
4	1.05	0.14	0.20	0.34	0.25	0.39	0.68	0.38	0.61	0.36	0.27	0.43
5	1.07	0.14	0.20	0.29	0.22	0.35	0.59	0.35	0.55	0.31	0.23	0.37
6	1.49	0.16	0.24	0.26	0.20	0.31	0.52	0.31	0.49	0.29	0.22	0.35
7	1.32	0.15	0.22	0.24	0.18	0.29	0.45	0.26	0.41	0.27	0.20	0.33
8	0.92	0.13	0.21	0.21	0.16	0.26	0.40	0.23	0.36	0.24	0.18	0.29
9	0.96	0.13	0.21	0.20	0.15	0.24	0.40	0.22	0.34	0.23	0.17	0.27
10	0.97	0.14	0.22	0.21	0.15	0.25	0.43	0.20	0.31	0.22	0.16	0.26
11	0.48	0.11	0.15	0.18	0.13	0.21	0.24	0.16	0.23	0.20	0.15	0.24
12	0.46	0.11	0.15	0.14	0.11	0.17	0.20	0.13	0.19	0.16	0.12	0.19
13	0.45	0.11	0.15	0.11	0.08	0.12	0.14	0.10	0.14	0.12	0.09	0.14

**Table 7 sensors-25-04073-t007:** Comparative summary of the model complexity and efficiency.

Model	Parameter Count	Training Time (s/Epoch)	Inference Speed (s/Sample)
U-Net	10,825,544	49.5	0.216
U-Net with SA	8,010,673	50	0.116
TS-MSDA U-Net	37,984,700	104	0.035
UNETR	57,424,712	59	0.032
UNETR++	8,783,600	105	0.111

## Data Availability

Data are available upon request.

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
