# Peer review of "U-Net Inspired Transformer Architecture for Multivariate Time Series Synthesis"

_sensors, 2025, doi:10.3390/s25134073_

Round 1
Reviewer 1 Report
Comments and Suggestions for Authors
This manuscript introduces TS-MSDA U-Net, a multiscale dual-attention U-Net architecture for multivariate time-series synthesis, validated on EV data and periodic signal reconstruction in power electronics. I recommend minor revision.
(1)Page 3, Line 130-136 The contributions related to the dual attention mechanisms would benefit from further quantification; the marginal gains in some experiments should be explicitly detailed to avoid overselling the novelty.
(2)Page 4, Line 146-164 The architectural description does not provide full hyperparameter details (e.g., layer numbers, filter sizes, activation types) or ablation study results, limiting reproducibility.
(3)Page 5, Line 197-202 The positional embedding method is described only briefly; specify whether sinusoidal, learned, or other embeddings are used, and clarify their impact via experiment or citation.
(4)Page 7, Line 233-246 The training protocols (number of epochs, learning rate schedules, and early stopping criteria) are insufficiently described for others to reproduce the results precisely.
(5)Page 8, Line 262-274 Dataset preprocessing omits details such as the handling of missing values, feature selection rationale, and window overlap percentage, which are critical for replicability.
(6)Page 10, Line 352-364 The discussion overstates the superiority of TS-MSDA U-Net over other attention models; please moderate the language and present statistical significance or effect sizes.
(7)Page 13, Line 380-391 The violin plot analysis does not include quantitative statistical tests (e.g., t-test, Wilcoxon) to support claims of "more balanced error distribution" across models.
(8)Page 15, Line 446-464 For the CLLC converter experiment, the relationship between simulation and real hardware data is not systematically discussed, especially concerning domain shift and potential overfitting to simulated conditions.
(9)Page 17, Line 488-517 The model comparison lacks an explicit ablation analysis to isolate the effect of each architectural modification (multiscale, dual attention, shared projections, etc.).
(10)Page 21, Line 579-583 Table 2 includes performance metrics, but no per-signal or per-component breakdown (e.g., Vds, Vc1, etc.) is provided, which would help in understanding where the model excels or underperforms.
(11)Page 22, Line 629-634 The conclusion refers to generalization capability in "domains characterized by nonlinear dynamics," but the EV and converter tasks are not sufficiently diverse to generalize this assertion? Please clarify & revise.
(12)For references, more related articles may be added: Geometry‐Aware 3D Point Cloud Learning for Precise Cutting‐Point Detection in Unstructured Field Environments; Journal of Field Robotics. Dual-Frequency LiDAR for Compressed Sensing 3D Imaging Based on All-Phase Fast Fourier Transform. J. Opt. Photonics Res.
Reviewer 2 Report
Comments and Suggestions for Authors
The paper considers the TS-MSDA U-Net model (a hybrid U-Net architecture complemented by bidirectional attention blocks) for the synthesis of long multidimensional time series. The model was tested in two tasks - synthesis of key parameters of an electric vehicle and reconstruction of high-frequency signals of a resonant CLLC converter. It is revealed that the modified model performs better in difficult conditions and scenarios, significantly surpassing the synthesis capabilities of the original model.
The article is quite well structured, contains a large amount of graphic material and summarizing tables. The theoretical material and the model are described in great detail, all the main modifications are discussed in detail with a presentation of graphical diagrams of architectures and blocks. The sources presented in the work are generally modern and relevant.
After reading it, I have several questions that require clarification and clarification:
1. I didn't really understand the highlighted area of the logarithmic scale in Figure 3. This area is not considered or described anywhere in the text. If it was meant that the right Y-axis has a logarithmic scale, then it was worth placing the signature there. Do all graphs drawn with dots belong to a logarithmic scale? In general, this point should be improved for a better understanding.
2. Is the error expressed as a percentage in Figure 4?
3. In Figure 5, it is also worth noting what the red borders mean. As I understand it, these are excluded tracks from consideration, but the caption to the picture could be expanded.
4. I think it's more at the discretion of the editors, but I rarely saw sections of highlights and conclusions before the abstract. They were usually placed at the end of the Introduction section.
5. The authors noted that the attention mechanisms did not provide much gain in the first task, but they played a crucial role in the second task. Visually, this is confirmed by the graphs, but have statistical tests been done on the differences between MAE/DTW of different models? Technically, this is a fairly simple procedure, and the tables could be supplemented with p-value values.
Thus, it is recommended to accept this work after minor revisions, taking into account the above comments.
Reviewer 3 Report
Comments and Suggestions for Authors
- The proposed TS-MSDA U-Net is primarily a combination of known techniques: U-Net backbone, dual attention mechanisms, and transformer-inspired components. The novelty is incremental rather than fundamental. As acknowledged by the authors themselves, the dual attention (DA) module offers only marginal gains in performance compared to the basic U-Net (e.g., Figure 3 and Section 3.1.1), which weakens the claimed innovation.
- No quantitative ablation is provided to isolate the effects of the sequence attention (SA) and channel attention (CA) modules separately. This prevents clear assessment of each module's utility. There is no comparison with simpler attention variants or efficient alternatives like lightweight attention or squeeze-and-excitation mechanisms.
- Equation formatting lacks clarity (e.g., boldface, matrix dimensions not consistently defined), which makes the methodology harder to follow.
-
Redundancy and repetition: Several points are repeatedly emphasized without adding new insights, especially in the abstract and introduction. Minor grammatical errors: For example, “The SA module aims is designed to model...” (Section 2.2.2). Ambiguous terms: Terms like “enhanced alignment to original data distributions” and “synthesized signals within ±1%” are vague unless accompanied by formal statistical measures.
- The model introduces attention modules and high channel dimensionality but does not report training time, parameter count, or inference speed.
-
Some works about dual attention mechanisms should be cited in this paper to make this submission more comprehensive, such as 10.1109/TPAMI.2024.3511621.
I recommend major revisions to enhance the quality of this manuscript. Additional details and explanations would greatly improve the manuscript.
Round 2
Reviewer 3 Report
Comments and Suggestions for Authors
I have thoroughly reviewed the revised version of the manuscript along with the authors' responses. It is evident that the authors have carefully considered and effectively addressed all the concerns that were previously raised. Therefore, I recommend accepting this manuscript
Author Response
Comment 1: I have thoroughly reviewed the revised version of the manuscript along with the authors' responses. It is evident that the authors have carefully considered and effectively addressed all the concerns that were previously raised. Therefore, I recommend accepting this manuscript
Response 1:
Thank you for your thorough review and positive feedback on our revised manuscript. We are delighted to hear that we have successfully addressed all of your previous concerns.
We truly appreciate your time and constructive comments, which have significantly helped us improve the manuscript.